# The Preventive Effect of A Magnetic Nanoparticle-Modified Root Canal Sealer on Persistent Apical Periodontitis

**DOI:** 10.3390/ijms232113137

**Published:** 2022-10-28

**Authors:** Xiao Guo, Yan Sun, Zheng Wang, Biao Ren, Hockin H. K. Xu, Xian Peng, Mingyun Li, Suping Wang, Haohao Wang, Yao Wu, Michael D. Weir, Xuedong Zhou, Fang Lan, Lei Cheng

**Affiliations:** 1State Key Laboratory of Oral Diseases, National Clinical Research Center for Oral Diseases, West China Hospital of Stomatology, Sichuan University, Chengdu 610041, China; 2Department of Cariology and Endodontics, West China Hospital of Stomatology, Sichuan University, Chengdu 610041, China; 3Department of Advanced Oral Sciences and Therapeutics, School of Dentistry, University of Maryland, Baltimore, MD 21201, USA; 4Stomatology Center, The First Affiliated Hospital of Zhengzhou University, Zhengzhou 450052, China; 5National Engineering Research Center for Biomaterials, Sichuan University, Chengdu 610064, China

**Keywords:** root canal sealer, persistent apical periodontitis, DMADDM, magnetic nanoparticles, biofilm

## Abstract

Persistent apical periodontitis is a critical challenge for endodontists. Developing root canal filling materials with continuous antibacterial effects and tightly sealed root canals are essential strategies to avoid the failure of root canal therapy and prevent persistent apical periodontitis. We modified the EndoREZ root canal sealer with the antibacterial material dimethylaminododecyl methacrylate (DMADDM) and magnetic nanoparticles (MNPs). The mechanical properties of the modified root canal sealer were tested. The biocompatibility of this sealer was verified in vitro and in vivo. Multispecies biofilms were constructed to assess the antibacterial effects of the modified root canal sealer. We applied magnetic fields and examined the extent of root canal sealer penetration in vitro and in vivo. The results showed that EndoREZ sealer containing 2.5% DMADDM and 1% MNP had biological safety and apical sealing ability. In addition, the modified sealer could increase the sealer penetration range and exert significant antibacterial effects on multispecies biofilms under an external magnetic field. According to the in vivo study, the apices of the root canals with the sealer containing 2.5% DMADDM and 1% MNP showed no significant resorption and exhibited only a slight increase in the periodontal ligament space, with a good inhibitory effect on persistent apical periodontitis.

## 1. Introduction

Bacterial infection is the major etiology of pulpitis and periapical periodontitis. Root canal treatment is the most effective therapy in clinical practice. However, infection in some teeth cannot be controlled after root canal treatment and develops into persistent apical periodontitis (PAP), resulting in pain, swelling, pulpitis, apical periodontitis, and even root resorption [1,2]. Two main factors cause the failure of root canal treatment. One is the failure to completely remove both the planktonic bacteria and biofilms in the root canal. For example, *Enterococcus faecalis* (*E. faecalis*) is abundant in apical periapical periodontitis and is a significant pathogen involved in PAP. *E. faecalis* can invade and grow in dentinal tubules [3,4]. Secondly, the anatomy of the root canal system is extremely complex, including accessory root canals, apical bifurcation, and C-shaped root canals [5,6]. As an important step in root canal treatment, root canal obturation plays a key role in sealing root canals and preventing bacterial reinfection. However, microleakage between the root canal sealer, root canal wall, and dentinal tubules can cause bacterial reinfection, leading to the failure of root canal treatment. Thus, the optimal root canal sealer should be able to penetrate the dentinal tubules and tightly fill and seal root canals from all angles. The deeper penetration of root canal sealer into dentinal tubules could effectively inhibit bacterial growth [7,8]. This also increases the surface contact area between the root canal filling materials and the root canal wall, improves the adaptability of the root canal wall and the root canal, and fills the root canal in three dimensions [9,10].

EndoREZ is a dual-cured methacrylate resin-based root canal sealer widely used in the clinic, and its hydrophilicity enhances its penetration into the dentinal tubules [11]. However, EndoREZ may pull resin sealer tags out of the dentinal tubules during polymerization shrinkage, creating gaps along the sealer—dentin interface, which may account for the reported suboptimal sealing of the EndoREZ system [11,12,13]. Dimethylaminododecyl methacrylate (DMADDM) is an antibacterial monomer of a quaternary ammonium salt. In previous studies, DMADDM was successfully synthesized and incorporated into different dental materials [14,15,16]. EndoREZ with a mass fraction of 2.5% DMADDM showed a significant antibacterial effect on multispecies biofilms without reducing its mechanical properties [17]. DMADDM has a double-bond structure that can bind to resin-based dental materials and be stably combined with the material to exert antibacterial effects. Meanwhile, Magnetic nanoparticles (MNPs) have been applied to magnetic resonance imaging (MRI), targeted drug delivery, gene therapy, tumor photothermal therapy, and biological separation [18,19]. MNPs can respond to magnetic fields, enabling the transport of more drugs to a target than via either diffusion or iontophoresis [20,21]. In our previous study, we successfully constructed a beagle-dog model of PAP more suited for evaluating the properties of dental materials compared with the rat models used in previous studies [22]. In this study, we modified the resin-based root canal sealer EndoREZ with DMADDM and MNPs. We hypothesized that the modified sealer could improve the penetrability of EndoREZ into dentinal tubes under a magnetic field to kill bacteria embedded in the deeper dentinal tubules in vitro and in vivo. In addition, the modified sealer might have good biocompatibility and promote apical periapical healing.

## 2. Results

### 2.1. Cytotoxicity and Material Properties

The schematic illustration of root canal sealer application in periapical model was shown as Figure 1. EndoREZ impregnated with different concentrations of DMADDM and MNP did not exhibit significant toxicity to L929 mouse fibroblasts, and the cell survival rate of each group was >75% (Figure 2A). As shown in Figure 2B, the solubility of the sealer containing 2.5% DMADDM + 2% MNP was significantly higher than the other groups (*p* < 0.05), with no significant difference in solubility between the other groups. The solubility of all the groups was <3%, following the ISO international standard for root canal sealers. The root canal sealer containing 2.5% DMADDM + 2% MNP exhibited decreased solubility (Figure 2C,D). Altogether, the results showed that the group with 2.5% DMADDM + 1% MNP had good biosafety and material properties compared to the control group.

### 2.2. The Penetration Range of the Sealer

In the coronal root, the penetration range of the sealer reinforced with MNP and DMADDM was significantly higher than that of the other four groups (*p* < 0.05; Figure 3A). In the middle root area, the penetration range of the sealers with 2.5% DMADDM + 1% MNP and 2.5% DMADDM + 2% MNP was higher than that of the control group (*p* < 0.05; Figure 3B). In the apical root area, the penetration range of the sealers with 2.5% DMADDM + 1% MNP and 2.5% DMADDM + 2% MNP was not statistically different from that of the control group (*p* > 0.05; Figure 3C). In the coronal and apical root areas, the penetration range and the depth of dye penetration of sealers with 2.5% DMADDM + 1% MNP and 2.5% DMADDM + 2% MNP were significantly higher than those of other groups.

### 2.3. Antibacterial Effects on Multispecies Biofilms

Compared with the EndoREZ group, the colony-forming units (CFU) counts in the sealer containing 2.5% DMADDM and MNP decreased significantly (*p* < 0.05; Figure 4A), and the antibacterial effect of the sealer impregnated with both DMADDM and MNP was the same as the sealer containing only DMADDM. Compared with the control group, the biomass of bacteria in the 2.5% DMADDM group decreased significantly (*p* < 0.05; Figure 4B). Meanwhile, the biofilms in the DMADDM group were much thinner and looser than in the other three groups (Figure 4D,E). Live/dead staining confocal images of biofilms on sealers were evaluated after 48 h. Live bacteria were stained green and dead bacteria were stained red (Figure 4E). The ratio of live/dead cells decreased in sealers containing DMADDM (Figure 4C). In the sealer group containing both 2.5% DMADDM and MNP, the proportion of *E. faecalis* decreased to a certain extent, and the proportion of L. acidophilus increased (Figure 4D).

### 2.4. Radiolucent Zones in the Periapical Region

In the present study, radiolucent zones in the periapical region and histopathological findings were used to assess the degree of inflammation. Four weeks after root canal exposure, all the root canals exhibited AP, with an average shadow volume of 6.04 ± 1.22 mm^3^ in the apical area. After root canal preparation and disinfection, the periapical shadow volume was 6.01 ± 1.05 mm^3^, which was not significantly different from the initial apical periodontitis group (*p* > 0.05), indicating that the treatment controlled the infection effectively. To establish PAP models, *E. faecalis* and the other test multispecies were inoculated into the root canals after treatment. The reinfection of the pathogen increased the area of the periapical shadow volume to 9.83 ± 1.21 mm^3^ (*E. faecalis* group) and 10.17 ± 1.18 mm^3^ (multispecies group), respectively (Figure 5). It was certified that PAP models induced by *E. faecalis* and multispecies strains were established successfully. In addition, there were no differences in periapical destruction between the *E. faecalis* and multispecies groups.

Next, we investigated the effect of DMADDM and MNP on these two PAP models. The change of the volume in radiolucent zones was 2.75 ± 1.89 mm^3^ for the EndoREZ group and 2.22 ± 1.15 mm^3^ for the EndoREZ + MNP group in *E. faecalis*-induced PAP, while 2.62 ± 1.87 mm^3^ for the EndoREZ group and 1.07 ± 1.47 mm^3^ for the EndoREZ + MNP group in multispecies-induced PAP, indicating that the periapical lesions continued to expand in the absence of antibacterial agents after root canal retreatment. In contrast, the DMADDM-modified sealer significantly reduced the volume of periapical lesions by 1.75 ± 0.69 mm^3^ and 1.87 ± 0.56 mm^3^ in the two PAP models, respectively. Moreover, the sizes of the lesions decreased more significantly when DMADDM and MNP were used in combination; further reductions of 2.85 ± 0.79 mm^3^ and 2.64 ± 0.56 mm^3^ being observed. Moreover, no significant difference was observed between the two PAP models.

### 2.5. Histopathological Findings

The histopathological findings are presented in Figure 6A. All the groups contained specimens with mineralized tissue resorption in different sizes. There was no apical mineralized tissue resorption in seven specimens in the EndoREZ + MNP + DMADDM group, while mineralized tissue resorption occurred in >50% of the specimens in the remaining three groups. Concerning the inflammatory infiltrate, a significant difference was observed between the EndoREZ group, the EndoREZ + DMADDM group, and the EndoREZ + MNP + DMADDM group. 80% of the specimens in the EndoREZ group showed severe inflammatory infiltrate. Furthermore, four specimens in the EndoREZ + MNP + DMADDM group were normal or exhibited a mild inflammatory infiltrate, with both being less than that in the EndoREZ + DMADDM group. The periodontal ligament thickness corresponded with the inflammatory infiltrate in each group. The EndoREZ and EndoREZ + MNP groups displayed moderately or severely increased periodontal ligament space, while 80% of the specimens in the EndoREZ + MNP + DMADDM group exhibited normal, or slightly increased, periodontal ligament space.

The representative histological images indicated that in the EndoREZ group, there was widespread inflammatory infiltration in the periapical area, and the resorption of the periapical mineralized tissues was severe. The image of the EndoREZ + MNP group was similar to that in the EndoREZ group, which showed severe apical bone resorption, with infiltration by many chronic inflammatory cells spreading to the surrounding area to form several microabscesses. In contrast, the scope of apical inflammation in the EndoREZ + DMADDM group was significantly lower than in the non-antibacterial modification groups, in which inflammatory cells were limited to the apical foramen, and neovascularization was seen in the periodontal ligament. Moreover, with the incorporation of MNP, the degree of inflammation further decreased, the periodontal ligament space increased slightly, and apical resorption was not observed (Figure 6B).

The immunohistochemistry results showed that many MMP-9-positive cells accumulated in the inflamed periapical tissue in the EndoREZ group, and the MMP-9 expression profile of the EndoREZ + MNP group was similar to that of the EndoREZ group. However, the expression of MMP-9 in the EndoREZ + DMADDM group was lower than in the EndoREZ group. Furthermore, a more pronounced decrease in MMP-9 expression was noticed in the EndoREZ + MNP + DMADDM group (Figure 6C). A similar tendency was observed concerning the expression of TNF-α in both the EndoREZ and EndoREZ + MNP groups, in which the TNF-α-positive cells in the periapical tissue exhibited moderate to intense staining. However, few positive cells were seen in the EndoREZ + DMADDM and EndoREZ + MNP + DMADDM groups (Figure 6D). In conclusion, our findings indicated that MNP could further enhance the antibacterial ability of DMADDM-modified sealer in vivo.

### 2.6. Antibacterial Effects and the Penetration Range of the Sealer In Vivo

To further explore the reason why the antibacterial effect of the EndoREZ + MNP + DMADDM group was better than that of the other three groups, we determined the penetration rate of the sealer under CLSM and analyzed microbiota of the periapical tissues by 16S rRNA sequencing. As shown in Figure 7A,B, in the coronal and apical root canal areas, the penetration rate of the EndoREZ group was significantly lower than that of the EndoREZ + MNP and EndoREZ + MNP + DMADDM groups (*p* < 0.05). In addition, there was no significant difference between the EndoREZ and EndoREZ + DMADDM groups. However, in the middle root area, although the penetration rate of the sealer impregnated with MNP was higher than that of the EndoREZ group, there was no significant difference between the EndoREZ group and the other three groups.

The results of 16S rRNA gene sequencing indicated various species in all the groups, and MNP and DMADDM significantly changed the microbial composition and relative abundance of periapical tissues. According to Figure 7C, there was a mean of 21.67 and 18 taxa in the EndoREZ and EndoREZ + MNP groups at the genus level, respectively. In contrast, there were 11.75 and 9 taxa in the EndoREZ + DMADDM and EndoREZ + MNP + DMADDM groups, respectively. Moreover, 9.67 and 7 taxa were identified in the EndoREZ and EndoREZ + MNP groups at the species level, respectively, which were higher than that in the EndoREZ + DMADDM (6.25) and EndoREZ + MNP + DMADDM (4) groups. Burkholderia occupied large proportions of the bacterial communities in all the groups. In addition, the dominant genera included Pseudomonas (9.6%) and Acinetobacter (3.5%) in the EndoREZ group, Prevotella (9.4%), Treponema (6%), and Fretibacterium (4%) in the EndoREZ + MNP group, Streptococcus (7.9%) and Lactobacillus (4.4%) in the EndoREZ + DMADDM group, and Fusobacterium (6.3%) and Muribaculum (2.8%) in the EndoREZ + MNP + DMADDM group (Figure 7D). Significantly, at the species level, the abundance of *E. faecalis* was 1.1% in the EndoREZ group. However, it was not detected in the other three groups (Figure 7E).

## 3. Discussion

As a filling material between the gutta-percha and the root canal wall, the root canal sealer dissolves over time, leaving a gap between the gutta-percha and the root canal wall and resulting in the failure of root canal treatment [23,24]. The results showed that the solubility of the sealer containing 2.5% DMADDM and 2% MNP increased compared with the other groups. The solubility in all the groups was <3%, conforming to ISO 6876 [25]. The apical sealing ability determines the outcome of root canal therapy [11,26]. The results showed that the apical sealing ability of the sealer reinforced with 2.5% DMADDM and 2% MNP decreased because of the dissolution of the sealer. Root canal sealers contact periapical tissues at the apical foramen; therefore, the cytotoxicity of the sealer is significant to the health of apical tissues [27,28]. The results showed that incorporating DMADDM and MNP had no obvious cytotoxicity, indicating that the modified sealer was still biologically safe.

Because of the complexity of the root canal system, it is necessary to use a sealer to penetrate the dentinal tubules [29,30]. There are some methods to investigate the dentinal tubule penetration of sealers, such as CLSM and SEM [31]. We choose CLSM to observe the sealer penetration. The results showed that the root canal sealer reinforced with MNP alone did not increase the sealer penetration in different parts compared with the control group. However, the 2.5% DMADDM + 1% MNP and 2.5% DMADDM + 2% MNP groups could penetrate the dentinal tubules more widely and deeply under the action of an external magnetic field, indicating that the antibacterial sealer could deeply penetrate the dentinal tubules and kill the bacteria located deep in the dentinal tubules. However, the solubility of the 2.5% DMADDM + 2% MNP group was significantly higher than the control group. Hence, incorporating 2.5% DMADDM and 1% MNP into the sealer could not only preserve the physical properties of the materials but also improve the penetration of the sealer so that it could penetrate the dentinal tubules more widely and deeply.

Developments in microbiology revealed that PAP was caused by infection with multiple strains, with *E. faecalis* as the dominant pathogen. Previous studies have demonstrated that *E. faecalis*, *S. gordonii*, *L. acidophilus*, and *A. naeslundii* isolated from root canals after treatment could form stable and repeatable multispecies biofilms in vitro [32]. Moreover, the multispecies model can consider the influence of materials on strain composition and proportion. Therefore, *E. faecalis*, *S. gordonii*, *L. acidophilus*, and *A. naeslundii* were selected to construct a multispecies biofilm model in vitro. This experiment investigated the antibacterial effect of root canal sealer modified by DMADDM and MNP on multispecies biofilms. The root canal sealer with 2.5% DMADDM could inhibit the multispecies biofilm; however, adding different concentrations of MNP had no apparent effect on the antibacterial properties of the root canal sealer. With the increase in MNP concentration, the antibacterial property of DMADDM was not affected. The observation of live/dead bacteria showed that the biofilms in the group reinforced with DMADDM were thinner and looser. The results of real-time quantitative PCR showed that the proportion of *E. faecalis* decreased in the group containing 2.5% DMADDM, indicating that DMADDM had a broad bactericidal effect and could reduce the proportion of *E. faecalis* to a certain extent in the multispecies biofilms.

Our present study used beagle dogs for PAP models by a two-step infection procedure adopted from previous studies [22,33]. Our previous research indicated that a DMADDM-modified sealer significantly reduced the inflammatory grade of periapical lesions. However, the antibacterial effect of the sealer containing DMADDM on the multispecies-induced PAP was not as good as that in the *E. faecalis*-induced reinfection [22]. We suspect that the enhanced proliferative capacity of *E. faecalis* in multispecies biofilms and increased extracellular matrix might make pathogens more resistant to DMADDM. Thus, enhancing penetration ability is required to improve antibacterial agents in root canal sealers. Our radiographic findings showed that incorporating MNP could further decrease the size of periapical lesions in both *E. faecalis*-induced and multispecies-induced PAP, with no significant differences between these two models. When the CLSM results were evaluated, it was suggested that MNP might allow the sealer to penetrate deep into the dentinal tubules or certain parts of the root canal system, such as lateral canals, apical ramifications, and furcation areas, resulting in the death of bacteria in the space where conventional sealers cannot penetrate. Due to the anatomical complexity of the root canal system, some pathogens persistently survive deep inside the dentinal tubules after chemo-mechanical treatment [2,34]. Many studies have demonstrated that bacteria can penetrate deeper than 1000 μm into the dentinal tubules under favorable conditions [35]. However, in a previous study, the mean tubular penetration depth of EndoREZ was found to be 448.9 μm at the coronal root, 359.6 μm at the middle root, and 360.4 μm at the apical root areas [36], which was not sufficient for obturating the living space for pathogens.

In the present study, TNF-α and MMP-9 were selected to evaluate the periapical inflammatory grade. The results indicated that the DMADDM significantly inhibited the expression of these two inflammatory markers, and the levels of TNF-α and MMP-9 further decreased due to the enhancement of MNP. TNF-α, as a cytokine, has been reported as a major contributor to bone loss [37]. A previous report indicated that *E. faecalis* strains isolated from the root canal system and saliva significantly up-regulated the expression of TNF-α in polymorphonuclear neutrophils (PMNs) [38]. In another study, dental pulp cells infected with *E. faecalis* demonstrated the greatest increase in TNF-α expression compared with other bacteria [39], consistent with the findings for the EndoREZ and EndoREZ + MNP groups in the present work. Matrix metalloproteinases (MMPs) family members are closely related to periapical inflammation [40,41]. MMP-9 is a gelatinase functioning in acute and chronic inflammatory diseases. During the periapical inflammatory process, the expression of MMP-9 is significantly higher than that in normal tissues [42,43], possibly explaining why there were more periapical mineralized tissues and neovascularization of the EndoREZ + MNP + DMADDM group in the histologic images because the expression of MMP-9 was present in osteoclasts and related to the number of neutrophils in inflammatory zones [44].

The persistent intraradicular infection caused by pathogens is the main factor of persistent AP [2]. Previous studies have identified approximately 100 different bacterial species residing in the root canal system after endodontic treatment [45,46]. Our findings showed that the EndoREZ group possessed the most taxa, indicating that the bacterial diversity of periapical lesions decreased with the penetration of MNP or due to the antibacterial effect of DMADDM. In the present study, periapical samples exhibited a diverse microbial profile, and *Burkholderia* predominated in all the groups, similar to a previous report on periradicular lesions [47]. Significantly, *E. faecalis* was only detected in the EndoREZ group. *E. faecalis* is the most frequently detected microorganism from PAP because of its ability to survive in harsh environments [48]. In addition, previous studies have demonstrated that *E. faecalis* has a stronger penetration ability than other species [49,50], which might be attributed to its virulence factors such as the biofilm-associated surface protein Esp and gelatinase GelE [51,52]. There are other potential alternatives to design effective antibacterial treatments against PAP. Fiber-reinforced composites (FRCs) used as dental materials show strong antimicrobial activity [53]. Moreover, the use of rod-shaped magnetic nanoparticles [54] can increase the load contact area and, thus, drive a more potent anti-fungal and antimicrobial response. Our results suggested that combining antibacterial materials and magnetic particles with excellent penetrating properties is a new technique to seal the root canal system where residual infection can persist and further eliminate the remaining microorganisms. It provides a new approach to preventing PAP.

## 4. Materials and Methods

### 4.1. Synthesis of DMADDM and Specimen Preparation

DMADDM was synthesized and verified according to a method described previously. The MNPs used consisted of ferrimagnetic magnetite (Fe_3_O_4_) particles 50–100 nm in size (Sigma-Aldrich, St. Louis, MO, USA) [55,56]. DMADDM and MNPs were mixed with EndoREZ (Ultradent, South Jordan, UT, USA) together or singly. The specimens were prepared according to a previous study [14].

These four groups were investigated:

(1) Unmodified EndoREZ (designated “Control”)

(2) EndoREZ + 1% MNPs (designated as “1% MNP”)

(3) EndoREZ + 2% MNPs (designated as “2% MNP”)

(4) EndoREZ + 2.5% DMADDM (designated as “2.5% DMADDM”)

(5) EndoREZ + 2.5% DMADDM + 1% MNPs (designated as “2.5% DMADDM + 1% MNP”)

(6) EndoREZ + 2.5% DMADDM + 2% MNPs (designated as “2.5% DMADDM + 1% MNP”)

The sealers were placed in cylindrical plastic molds (5 mm in diameter and 2.5 mm in height) for cytotoxicity and solubility tests. Resin flakes were made using a 48-well plate cover as a mold to test the antibacterial effect; 20 mg of each sealer was placed on the flakes as flatly as possible. All the samples were transferred into new 24-well plates in a humid environment at 37 °C for 7 days and sterilized using an ethylene oxide sterilizer.

### 4.2. Cytotoxicity Test 

The CCK-8 assay was used to determine the sealer’s cytotoxicity on the mouse fibroblast permanent cell line L929. Each specimen was immersed in 10 mL of Dulbecco’s Modified Eagle Medium (DMEM) for 24 h. The cells were seeded in a 96-well plate (5 × 10^3^ cells per well) and grown overnight. The cells were then incubated with elution for 24 h. Next, the cells were washed twice with PBS; 10 µL of CCK-8 reagent and 100 µL of fresh medium culture were added to each well. After incubation at 37 °C for 1 h, the solution’s absorbance was measured at 450 nm.

### 4.3. Evaluation of Solubility

The samples were prepared and weighed three times in each group and recorded as W1. The samples were immersed in 10 mL of distilled water for 14 days and then placed in a drying oven to dry completely. Each sample was weighed three times, and the weight was recorded as W2. The solubility (S) was calculated using the following formula: S = (W1 − W2)/W1 × 100%.

### 4.4. Root Canal Preparation and Apical Sealing Ability

Sixty single-rooted human anterior teeth with fully formed apices were anonymously collected from the hospital and stored in normal saline solution at 4 °C until use. The study was approved by the Ethical Committee of the West China School of Stomatology, Sichuan University (Chengdu, China, WCHSIRB-D-2017-165). The detailed methods are described in the Appendix A. After seven days, the teeth were treated and observed under a stereomicroscope as described in the Appendix A.

### 4.5. Root Sectioning and Confocal Laser Scanning Microscopy Analysis of the Roots

After seven days, the roots were embedded vertically in resin and sectioned horizontally at three levels (2, 5, and 8 mm from the root apex) using a water-cooled low-speed blade to obtain 1 ± 0.1 mm thick slices. The specimens were examined under a confocal laser scanning microscope, and the images of sections were obtained. The images were analyzed using Camera Measure (E2ESOFT, Shanghai, China). First, regions, where the sealer penetrated into the dentinal tubules along the root canal walls were measured. Next, this value was divided by the circumference of the root canal wall, which was multiplied by 100 to calculate the percentage.

### 4.6. Biofilm Formation 

*Enterococcus faecalis* ATCC29212, *Streptococcus gordonii* ATCC35105, *Lactobacillus acidophilus* ATCC4356, and *Actinomyces naeslundii* ATCC12104 were provided by the State Key Laboratory of Oral Diseases (Sichuan University, Chengdu, China). For multispecies biofilm formation, bacterial suspensions were mixed to obtain an inoculum containing a defined microbial population consisting of *E. faecalis* (1 × 10^6^ CFUs/mL), *S. gordonii* (1 × 10^6^ CFUs/mL), L. acidophilus (1 × 10^6^ CFUs/mL), and *A. naeslundii* (1 × 10^6^ CFUs/mL) in 2 mL of BHI in 24-well plates containing the specimens anaerobically for 48 h. The bacterial culture medium was changed every 24 h.

### 4.7. Colony-Forming Units (CFU) and Crystal Violet Assay

The biofilms were incubated in BHI medium for the Colony-forming units (CFU) and crystal violet assay. The detailed methods were described in Appendix A.

### 4.8. Biofilm Imaging

The specimens were treated and observed by scanning electronic microscopy and confocal laser scanning microscopy.

### 4.9. Animal Study

In the present study, an animal model was performed according to our previous description [22]. All the animal experiments were conducted strictly following the guidelines of the Ethics Committee of Sichuan University and the West China School of Stomatology (Chengdu, China, WCHSIRB-D-2017-114). 

Four beagle dogs (12 months, weighing 10 kg) were obtained from the animal resource center (Sichuan University, Chengdu, China). In total, 80 root canals of maxillary and mandibular premolars were randomly included. Preoperative radiographs of premolars were prepared using cone-beam computed tomography (CBCT) to ensure no inflammation in the periapical tissue. The beagle dogs were anesthetized by intraperitoneal injection of 3% pentobarbital sodium (30 mg/kg body weight), and the pulp chambers of the selected premolars were opened to expose the root canals to the oral environment for four weeks to induce chronic AP. The periapical lesions were confirmed by CBCT. Four weeks after pulp exposure, the root canals were shaped to #25 by S3 nickel-titanium rotary instruments with irrigation of 1.0% sodium hypochlorite solution. Then, the root canals were dried with sterile paper points and filled with calcium hydroxide, followed by access cavity sealing with an adhesive resin.

Two weeks after disinfection, CBCT was used to observe changes in the extent of the periapical lesion. Then, the root canals were equally divided into *E. faecalis* and multispecies groups. For the *E. faecalis* group, the root canals were inoculated with *E. faecalis* suspension (10^6^ CFUs/mL). For the multispecies group, the root canals were inoculated with a mixed bacterial suspension (*E. faecalis*, *L. acidophilus*, *A. naeslundii*, and *S. gordonii*; the concentration of all the strains was 10^6^ CFUs/mL). Finally, the cavity was sealed with an adhesive resin. An imaging evaluation of the periapical lesion was performed by CBCT two weeks later.

Next, the root canals were prepared to #35 and irrigated with 1.0% sodium hypochlorite solution. For root canal obturation, the root canals were divided into 4 groups: EndoREZ, EndoREZ + 1% MNP, EndoREZ + 2.5% DMADDM, and EndoREZ + 1% MNP + 2.5% DMADDM. After drying, the root canals were obturated with gutta-percha points and sealers as described above by the vertical condensation technique, followed by access cavity sealing with an adhesive resin. Postoperative periapical radiographs were taken after three months. In the present study, all CBCT images were performed with MIMICS V21.0 to calculate the shadow volume in the apical area.

### 4.10. Samples Collection and 16S rRNA Genetic Sequencing

Four root canals from the EndoREZ, EndoREZ + 1% MNP, EndoREZ + 2.5% DMADDM, and EndoREZ + 1% MNP + 2.5% DMADDM groups were randomly selected for microbial identification. In brief, after tooth extraction, the periapical lesions were collected in a sterile condition and sent to the Shanghai Personal Biotechnology Co., Ltd. (Shanghai, China), where the total DNA was extracted, amplified, and sequenced according to standard procedures.

### 4.11. Hematoxylin-Eosin (HE) Staining

The maxillae and mandibles of the beagle dogs were fixed in a 10% neutral-buffered formalin solution, dehydrated, and embedded in paraffin. After preparing 5–6-μm sections from the paraffin blocks and staining with HE, images were acquired using inverted light microscopy. The inflammation grade scoring criteria of all the specimens were based on a previous study [22], and the pathological evaluation was carried out from three aspects: inflammatory infiltration, the thickness of the periodontal ligament, and the resorption of mineralized tissues (Appendix A).

### 4.12. Immunohistochemical Staining

After dewaxing by xylene and hydration using graded alcohol, the sections were treated with citrate antigen retrieval solution to repair antigen at 80 °C for 20 min. The sections were treated with 3% H_2_O_2_ for 10 min and then blocked for 1 h. Diluted primary antibodies (TNF-α: 1:300, Novus, NBP1-19532 and MMP-9:1:200, Novus, NBP2-13173) were used to incubate the sections at 4 °C overnight. After washing with PBS solution three times (5 min each time), the tissue slices were incubated with a secondary antibody (1:200) and Streptavidin-HRP regent for 30 min at room temperature. Finally, the sections were treated with DAB substrate and hematoxylin.

### 4.13. Statistical Analysis

Each experiment was independently repeated at least three times. One-way analysis of variance (ANOVA) was performed to detect the significance of the variables. The Student Newman–Keuls test was used for all pairwise comparisons. Nonparametric Kruskal–Wallis analysis and the Mann–Whitney U-test were used to analyze the change of apical shadow volume and histopathologic results. Statistical analysis was performed with the SPSS software, version 23.0 (SPSS Inc., Chicago, IL, USA). Significant differences were considered when *p* < 0.05.

## 5. Conclusions

The root canal sealer modified by DMADDM and MNP exhibited excellent material properties and could improve the permeability of the sealer under the action of a magnetic field to penetrate deeper and wider into dentin tubules. It had a significant inhibitory effect on multispecies biofilms, with good clinical application prospects. In addition, it could promote apical periapical healing in PAP.

## Figures and Tables

**Figure 1 ijms-23-13137-f001:**
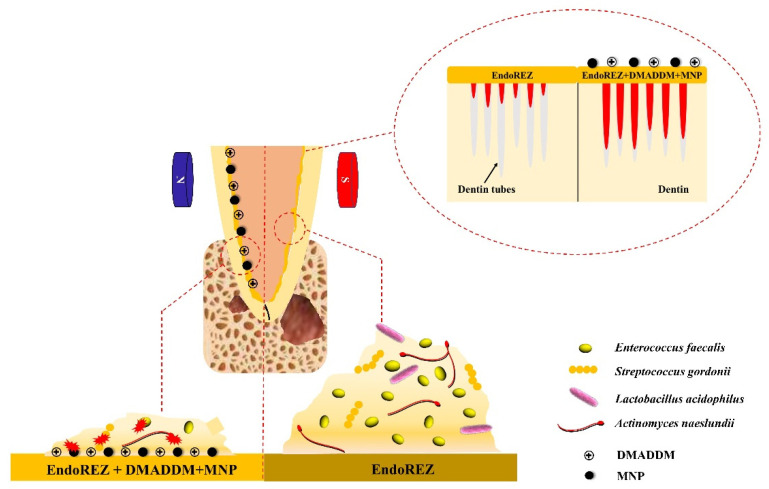
Schematic illustration of root canal sealer application in periapical model.

**Figure 2 ijms-23-13137-f002:**
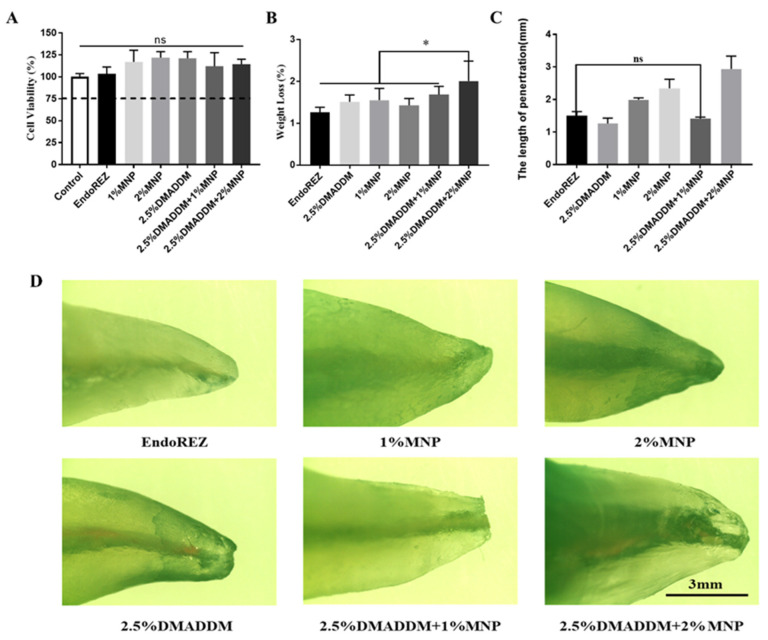
Correlation property tests of sealers. (**A**) Cytotoxicity assay of sealer eluents with mouse fibroblast (*n* = 5). (**B**) Solubility test of sealers in different groups (*n* = 5). (**C**) Apical sealing ability test of the sealers (*n* = 5). (**D**) Images of apical sealing ability recorded using a stereomicroscope. ns *p* > 0.05, not significant, * *p* < 0.05. Values are presented as means ± SD.

**Figure 3 ijms-23-13137-f003:**
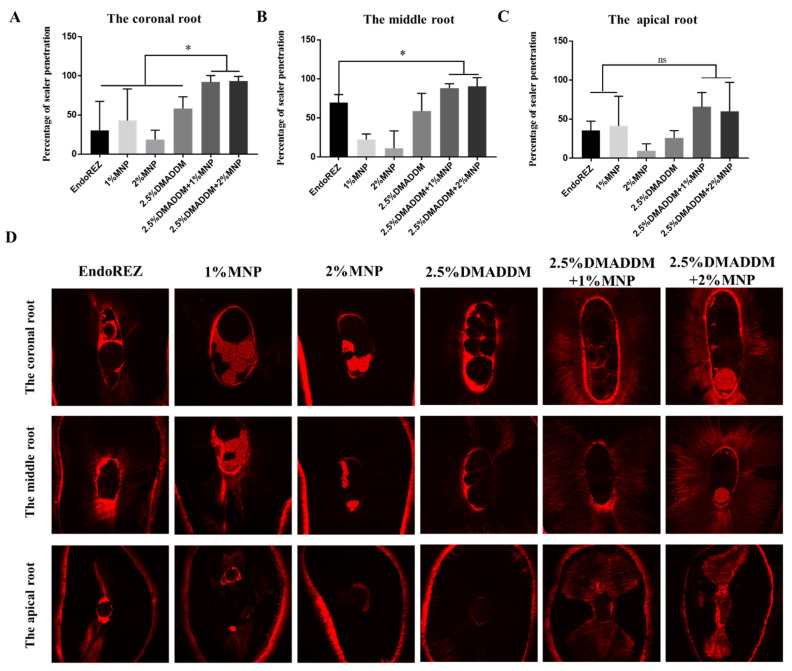
The results of percentage of penetration. Percentage of sealer penetration at the coronal root (**A**), middle root (**B**), and apical root (**C**). (**D**) Representative confocal laser scanning microscopy images from each experimental group at coronal, middle, and apical regions. (*n* = 5) ns *p* > 0.05, not significant, * *p* < 0.05. Values are presented as means ± SD.

**Figure 4 ijms-23-13137-f004:**
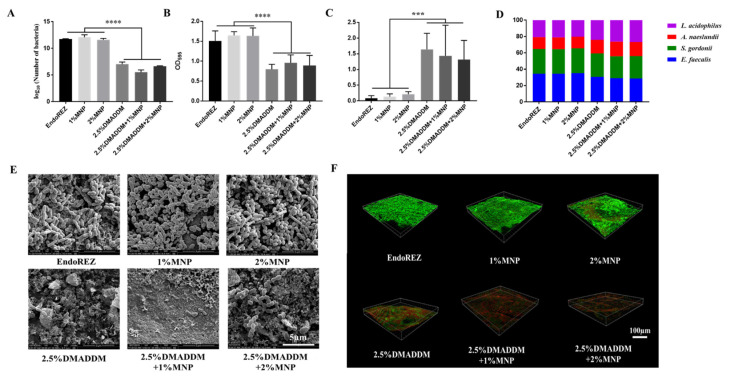
Antibacterial effect of the sealers on multispecies biofilms. (**A**) Colony-forming unit counts of biofilms (*n* = 5); (**B**) Biofilms biomass on different groups after 48 h, tested via crystal violet assay (*n* = 5); (**C**) The ratio of dead/live bacteria computed in line with three random sites of biofilms (*n* = 5); (**D**) Ratios of four bacteria species in multispecies biofilms formed on the sealers (*n* = 5); (**E**) SEM images of multispecies biofilms. (**F**) Live/Dead bacteria staining assay. *** *p* < 0.001, **** *p* < 0.0001. Values are presented as means ± SD.

**Figure 5 ijms-23-13137-f005:**
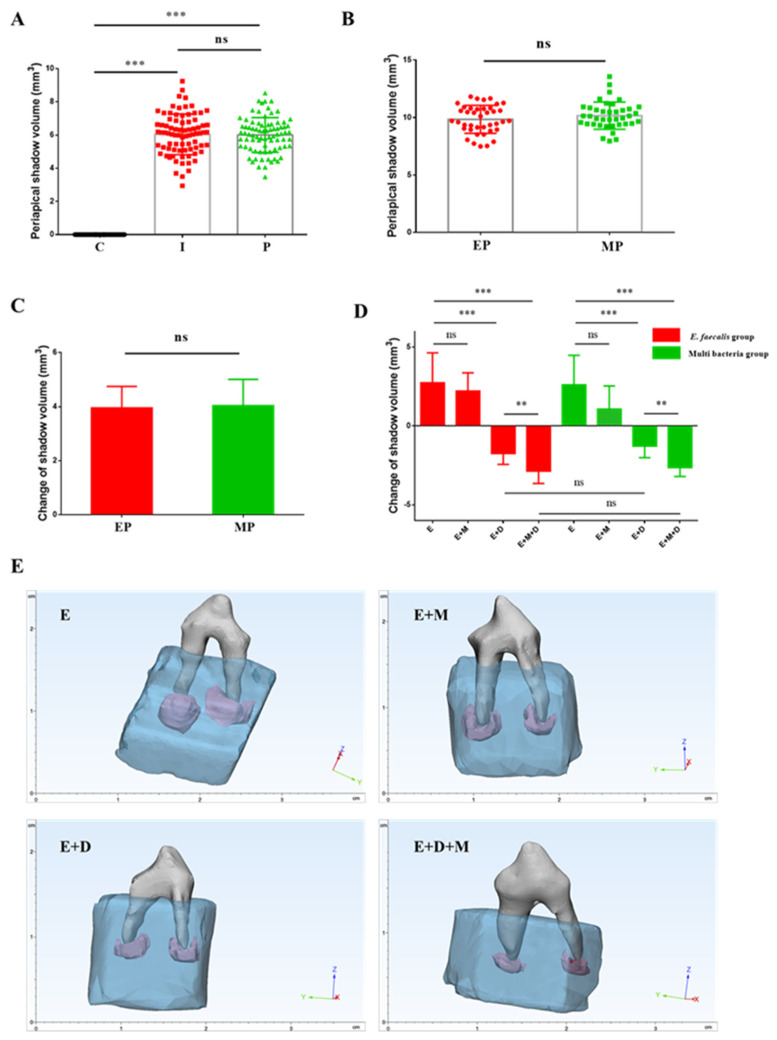
The volume of periapical radiolucent zones in each group (**A**) “C” represents the control group (before cavity preparation), “I” represents the initial chronic AP group and “P” represents the root canal preparation and disinfection group (*n* = 80); (**B**,**C**) “EP” represents *E. faecalis* induced persistent AP and “MP” represents multi-bacteria induced persistent AP (*n* = 40); (**D**) Changes of periapical radiolucent volume after root canal filling. “E” represents the EndoREZ group, “E + M” represents the EndoREZ + MNP group, “E + D” represents the EndoREZ + DMADDM group, “E + M + D” represents the EndoREZ + MNP + DMADDM group, and the same below (*n* = 10); (**E**) CBCT reconstruction by 3-matic Research 13.0 showing the representative periapical radiolucent area image of the premolar tooth. Teeth are shown in grey, alveolar bones are shown in blue, and periapical lesions are shown in violet. ns *p* > 0.05, not significant, *p* > 0.05, ** *p* < 0.01, *** *p* < 0.001.

**Figure 6 ijms-23-13137-f006:**
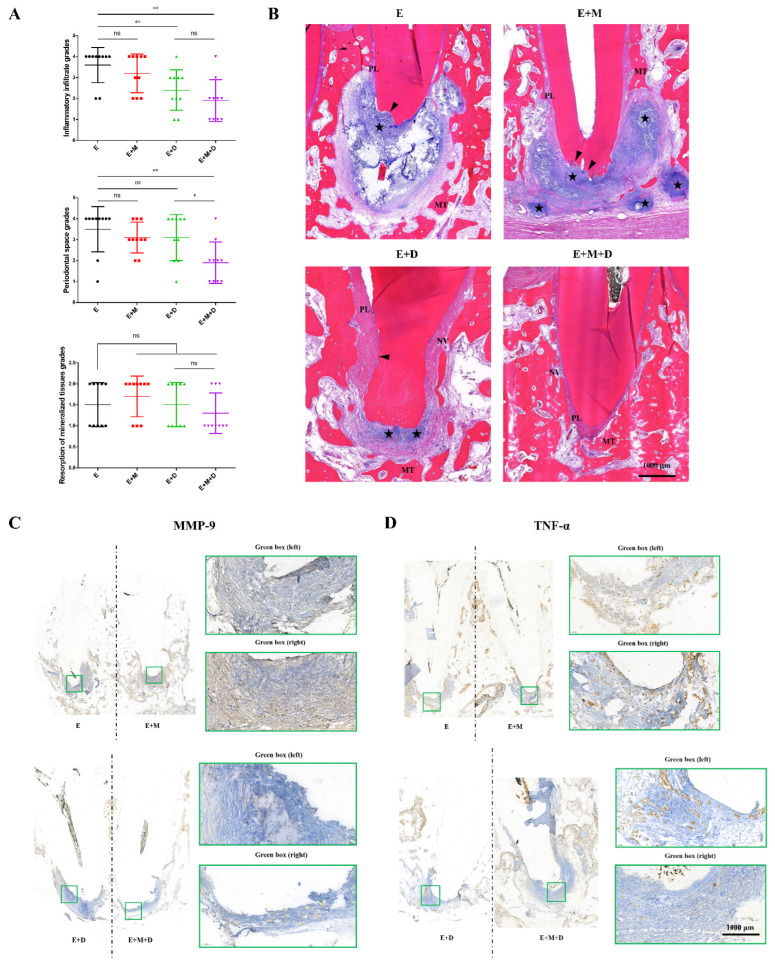
Histological and immunohistochemistry results of periapical lesions in each group. (**A**) Representative histopathological images in periapical areas after root canal filling; (**B**) The inflammation grade evaluation of periapical lesions (*n* = 10); (**C**) Representative immunohistochemistry (IHC) staining of MMP-9 in periapical areas after root canal filling. Green boxes indicate peripheral periodontal tissue; (**D**) Representative immunohistochemistry (IHC) staining of TNF-α in periapical areas after root canal filling. Green boxes indicate peripheral periodontal tissue. ns *p* > 0.05, not significant, * *p* < 0.05, ** *p* < 0.01.

**Figure 7 ijms-23-13137-f007:**
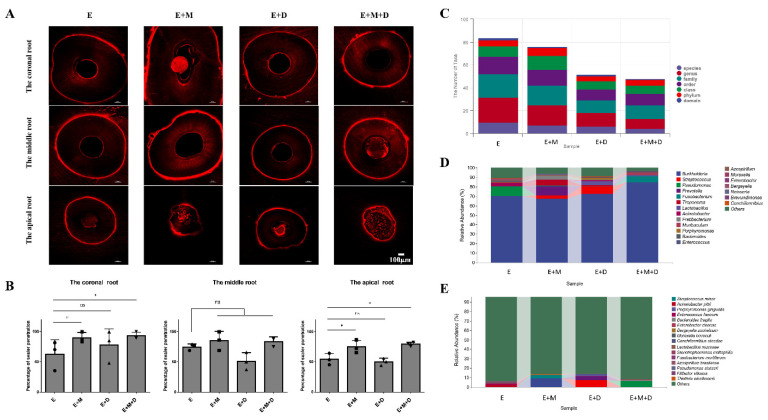
The results of percentage of sealer penetration and 16s rRNA sequence analysis in vivo. (**A**) Representative confocal laser scanning microscopy images from each group at the coronal, middle, and apical root regions; (**B**) Percentage of sealer penetration at the coronal root, middle root, and apical root (*n* = 3); (**C**) The number of taxa of bacteria from periapical tissues at each classification level (*n* = 4); (**D**) Abundance of bacteria from periapical tissues at genus level (*n* = 4); (**E**) Abundance of bacteria from periapical tissues at species level (*n* = 4). ns *p* > 0.05, not significant, * *p* < 0.05.

## Data Availability

The data that support the findings of this study are available from the corresponding author upon reasonable request.

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
