# Peer review of "The Preventive Effect of A Magnetic Nanoparticle-Modified Root Canal Sealer on Persistent Apical Periodontitis"

_ijms, 2022, doi:10.3390/ijms232113137_

Round 1

Reviewer 1 Report

Well written paper with very good results.

With this study, a new sealer is presented, which is presented in a previously unknown way (magnetic nano- particles) in its effect. Due to the novelty and the effectiveness, this study is already interesting for every practionioner. Furthermore, it is scientifically exact from the methodological structure and the presentation of the data.

Author Response

Dear editor and reviewers:

We thank you for all the constructive comments and suggestions on our manuscript entitled “The preventive effect of a magnetic nanoparticle-modified root canal sealer on persistent apical periodontitis”. These comments are all valuable and helpful for improving the paper. We have substantially revised our manuscript after reading the comments provided by the editor and reviewers. We look forward to hearing from you regarding our submission. We would be glad to respond to any further questions and comments that you may have.

The main corrections in the paper and the responses to the reviewer’s comments are as following file.

Reviewer 2 Report

The manuscript titled “The preventive effect of a magnetic nanoparticle-modified root canal sealer on persistent apical periodontitis” by Guo, X.; et al. is an original scientific work where the authors study the biocompatibility, the penetration length at coronal roots, the antibacterial response and the distribution of sealers with different content of dimethylaminodedocyl methacrylate (DMADDM) and magnetic nanoparticles (MNPs).  Authors found the conditions of EndoREZ sealer containing 2.5% DMADDM and 1% MNP the most promising for periodontitis biomedical purposes. The scientific approach and methodology followed by the authors seem right and the gathered results can be relevant for the examined field. The knowledge acquired in the present work could significantly aid in the design of more efficient treatments against bacterial infections in patients with periodontitis. The results achieved are well-discussed during the main body of the reported manuscript. The scientific paper is well written. In my opinion the present manuscript is innovative and the methodological approached used matches with the scope of International Journal of Molecular Sciences. For the above described reasons, I recommend the publication in International Journal of Molecular Sciences once the following remarks will be fixed:

--------

INTRODUCTION

Introduction section is clear and concise. No actions are required for this section.

--------

RESULTS

Authors perfectly state the most relevant outcomes found in the present work. Some points should be addressed to improve the manuscript quality.

I) Figure 3D, Figure 4E, Figure 6C and Figure 7A. Please, scale bars should be added in these Images.

II) Figure 2 (line 90). Authors indicate the population size (n): “(n=5)”. This information should be also available in the caption of the rest of Figures.

III) When authors explain the results gathered by the scanning electron microscopy (SEM) technique the following reference should be added to highlight the potential capabilities of this technique in this field [1].

[1] Patri, G.; Agrawal, P.; Anushree, N.; Arora, S.; Kunjappu, J.J.; Shamsuddin, S.V. A Scanning Electron Microscope Analysis of Sealing Potential and Marginal Adaptation of Different Root Canal Sealers to Dentin: An In Vitro study. J. Contemp. Dent. Pract. 2020, 21, 73-77.

IV) I miss information concerning the magnetic nanoparticle physical properties. Authors only provide information about their chemical nature in the respective Material and Methods section “MNPs were Fe3O4 nanoparticles (Sigma-Aldrich, USA)” (line 345). What is the size distribution of these nanoparticles? Did the authors conduct negative controls on this regard to check de dimensions delivered by the manufacturer? (e.g. by SEM, transmission electron microscopy (TEM) or dynamic light scattering (DLS)). In case affirmative, an additional figure should be prepared and placed in the respective Supplementary Information material. In case negative, at least some information should be furnished linked with relevant bibliography (Sigma usually affords some appropriate reference citations for each of their chemical products and consumables).

V) “(…), the CFU counts (…)” (line 109). Please, define “colony-forming units” term and place CFU between brackets.

VI) Data coming from Section “3.4. Radiolucent Zones in the Periapical Region” (lines 127-163) contain two significant figures (e.g. “The change of the volume in radiolucent zones was 2.75 ± 1.89 mm3 for the EndoREZ group and 2.22 ± 1.15 mm3 for (…)” (lines 142-143)), whereas the results from section “3.6. Antibacterial Effects and the Penetration Range of the Sealer in vivo” (lines 211-244) only exhibit one significant figure (e.g. “In addition, the dominant genera included Pseudomonas (9.6%) and Acinetobacter (3.5%) (…)” (lines231-232)). Please, authors should homogenize this point through the entire manuscript body text.

--------

DISCUSSION

Discussion is well structured. Authors should address potential alternatives to design effective antibacterial treatments against apical periodontitis. In this context, fiber-reinforced composites (FRCs) can meet these criteria [2]. Moreover, the use of rod-shape magnetic nanoparticles [3] can increase the load contact area and thus, driven more potent anti-fungal and antimicrobial response.

[2] Esteban-Tejeda, L.; Cabal, B.; Torrecillas, R.; Prado, C.; Fernandez-Garcia, E.; López-Piriz, R.; Quintero, F.; Pou, J.; Penide, J.; Moya, J.S. Antimicrobial activity of submicron glass fibres incorporated as a filler to a dental sealer. Biomed. Mater. 2016, 11, 015014. https://doi.org/10.1088/1748-6041/11/4/045014.

[3] Marcuello, C.; Chambel, L.; Rodrigues, M.S.; Ferreira, L.P; Cruz, M.M. Magnetotactic Bacteria: Magnetism Beyond Magnetosomes. IEEE Trans. Nanobioscience. 2018, 17, 555-559. https://doi.org/10.1109/TNB.2018.2878085.

--------

MATERIALS AND METHODS

MNPs were Fe3O4 nanoparticles (Sigma-Aldrich, USA)” (line 345). This sentence sounds redundant. I may strongly encourage the authors to modify it by “MNPs made by ferrimagnetic magnetite (Fe3O4) (Sigma-Aldrich, USA)”.

What are the suppliers of the material used in section “2.12 Immunohistochemical Staining” (lines 455-463). Please, authors should provide this information.

--------

CONCLUSIONS

This section perfectly remarks the most significant findings of the submitted manuscript. No further actions are required.

--------

REFERENCES

Bibliography citations are in the proper format of International Journal of Molecular Sciences. Authors only should take care of the journal page range (e.g. reference number 48 should be modified from “1207-13” to “1207-1213”. Check this point for the rest of citations.

--------

OVERVIEW AND FINAL COMMENTS

The submitted work is well-designed and the gathered results are interesting to devote the next-generation of antimicrobial therapies against periodontitis disease. For this reason, I will recommend the present scientific manuscript for further publication in International Journal of Molecular Sciences once all the aforementioned suggestions will be properly fixed.

PS: Please, note that I marked “major revisions” mainly due to the concern specified in point IV) in Results section.

Author Response

(The authors gave the same response as above.)

Round 2

Reviewer 2 Report

Authors fulfilled my previous request. For this reason, the quality of the scientific manuscript has been improved. I warmly suggest its further consideration to be published in IJMS journal